# Reinforced Pipeline Optimization: Behaving Optimally with Non-Differentiabilities

## Abstract

Many machine learning systems are implemented as pipelines. A pipeline is essentially a chain/network of information processing units. As information flows in and out and gradients *vice versa*, ideally, a pipeline can be trained end-to-end via backpropagation provided with the right supervision and loss function. However, this is often impossible in practice, because either the loss function itself may be non-differentiable, or there may exist some non-differentiable units. One popular way to superficially resolve this issue is to separate a pipeline into a set of differentiable sub-pipelines and train them with isolated loss functions. Yet, from a decision-theoretical point of view, this is equivalent to making myopic decisions using *ad hoc* heuristics along the pipeline while ignoring the real utility, which prevents the pipeline from behaving optimally. In this paper, we show that by converting a pipeline into a stochastic counterpart, it can then be trained end-to-end in the presence of non-differentiable parts. Thus, the resulting pipeline is optimal under certain conditions with respect to any criterion attached to it. In experiments, we apply the proposed approach - reinforced pipeline optimization - to Faster R-CNN, a state-of-the-art object detection pipeline, and obtain empirically near-optimal object detectors consistent with its base design in terms of mean average precision.

## 1 Introduction

In the context of *machine learning*, many systems are implemented as pipelines consisting of multiple stages, where each stage is essentially an information processing unit. Arranged along a chain or over a network, the output of previous stages constructs the input to the subsequent stages that are directly connected. With the success of *deep learning*, many stages within pipelines are nowadays implemented as neural networks, which are trainable via backpropagation (Werbos, 1988). Treating the whole pipeline as a large computation graph, just like what we do for a regular neural network, it would be desirable to train it end-to-end via backpropagation provided with the right supervision and loss function, since end-to-end training usually ensures optimality.

However, this is often impossible in practice, because a pipeline often contains some non-differentiable parts. Either the loss function itself may be non-differentiable, or there may exist some non-differentiable computation nodes. Take Faster R-CNN (Ren et al., 2015), a state-of-the-art object detection pipeline shown in Figure 1, as an example. It consists of a region proposal network (RPN) and a region convolutional neural network (R-CNN) sharing a single backbone network (Girshick, 2015). There are many non-differentiable parts within Faster R-CNN. The final evaluation criterion, i.e. mean average precision (mAP), is non-differentiable; the hard-coded region-of-interest (RoI) pooling and non-maximum suppression (NMS) operations are also non-differentiable with respect to the bounding box scores and coordinates. Such non-differentiabilities present significant challenges for training Faster R-CNN end-to-end. This is why the original implementation in (Ren et al., 2015) adopts a 4-step training strategy to approximate end-to-end learning.

The existence of non-differentiable parts prevents the whole pipeline from being trained end-to-end or optimized directly with respect to the final evaluation criterion. Prior approaches to training such pipelines rely on: 1) separate it into a set of isolated sub-pipelines where each sub-pipeline involves only differentiable parts; and 2) cherry-pick one appropriate differentiable loss function for each sub-pipeline to jointly mimic the final criterion. However, due to the interruption of the original in-

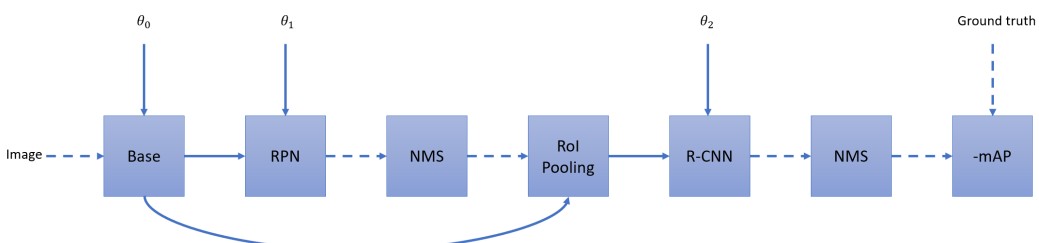

Figure 1: Faster R-CNN pipeline represented as a computation graph in testing time. Dotted lines indicate non-differentiabilities — either the edge itself is non-differentiable or the gradient passing on the edge is intentionally ignored. In testing time, network parameters $\theta_i$ are kept frozen. Noting that -mAP is the real criterion measuring the performance of this pipeline.

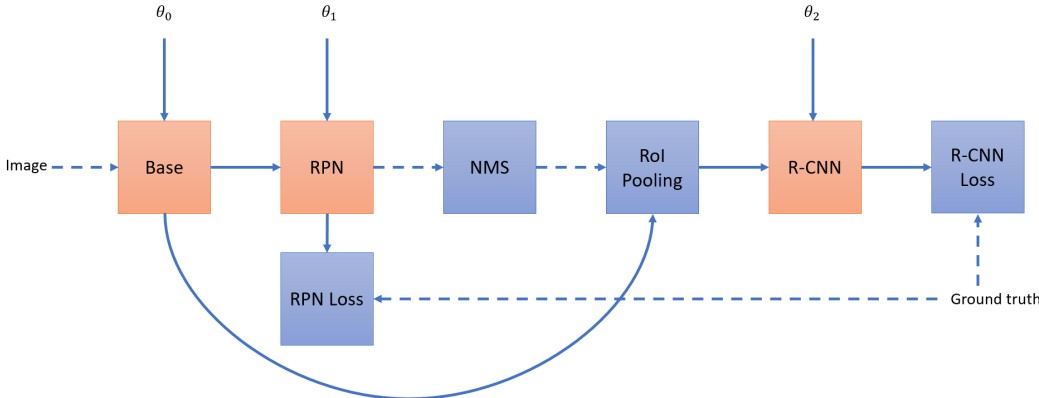

Figure 2: Traditional Faster R-CNN pipeline represented as a computation graph in training time compromised to non-differentiabilities. Nodes in orange color indicate learning models. Lacking the ability of optimizing the real criterion -mAP, isolated RPN and R-CNN losses are being optimized instead.

formation flow and the introduction of additional loss-function branches, the output distributions of previous sub-pipelines cannot be aligned with the subsequent sub-pipelines. Moreover, this causes inconsistency between training and testing, since different criteria are being used. As in the original implementation of Faster R-CNN, RPN and R-CNN are basically trained within their own loops with separate loss functions unaware of the other, although, after training, they have to work together awkwardly. Figure 2 shows the diagram of Faster R-CNN pipeline in training time following this training strategy. From a decision-theoretical point of view, training a pipeline in this way is equivalent to making myopic decisions using *ad hoc* heuristics for each sub-pipeline while completely ignoring the real utility — which is easily the opposite of *doing the right thing* (Russell & Wefald, 1991). As a result, no optimality can be ensured in any form for the pipeline either in training or testing time.

In this paper, according to the theory of *stochastic computation graph* (SCG) (Schulman et al., 2015), we show that by applying a so-called *stochastication* trick on an originally non-differentiable pipeline, we obtain a stochastic but differentiable counterpart which can then be trained completely end-to-end via backpropagation. The conversion from the original pipeline to the stochastic counterpart does not change the underlying information flow, or introduce any additional branches, thus the input- and output- distributions are always kept aligned along the pipeline. Furthermore, the gap between training and testing is closed since the same criterion is being used in both times.

We show that under certain conditions the stochastic pipeline and the original pipeline share the same set of optimal parameters. Since the final criterion is the only objective being optimized here, the learned pipeline is optimal over the training data. If the testing data is sampled from the same distribution as the training data — which is usually true in most benchmark dataset, the

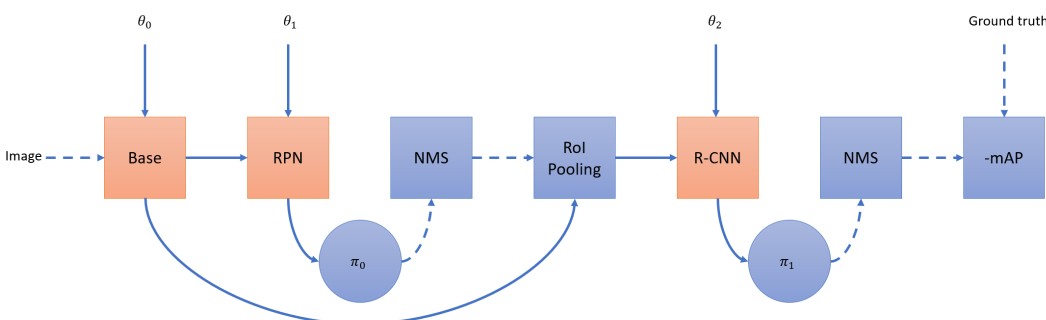

Figure 3: Reinforced Faster R-CNN pipeline represented as a SCG in training time, where $\pi_0$ and $\pi_1$ are the augmented stochastic nodes. This pipeline optimizes directly on the real criterion -mAP, and behaves the same way in both training and testing time, except that in testing time sampling is actually not needed, and parameters are kept frozen.

learned pipeline is also optimal over the testing data asymptotically. In experiments, we apply the proposed approach - reinforced pipeline optimization (RPO) - to Faster R-CNN, and obtain empirically near-optimal Faster R-CNN pipelines consistent with base models with respect to mAP. Figure 3 summarizes the main approach proposed in this paper.

The rest of this paper is organized as follows. In Section 2, we introduce the theory of stochastic computation graph, and present the basic method for doing backpropagation over SCGs. In Section 3, we formally define a machine learning pipeline, introduce the so-called stochastication trick, and present our main algorithm for end-to-end pipeline optimization. In Section 4, we show the main experimental results on optimizing a Faster R-CNN pipeline. In Section 5, we discuss some related work. And finally, in Section 6, we conclude with future work.

## 2 BACKGROUND

In this section, we define stochastic computation graph, and present the method of gradient estimation over stochastic computation graphs.

### 2.1 STOCHASTIC COMPUTATION GRAPH

Following the work of Schulman et al. (2015), we define stochastic computation graph as follows.

**Definition 1 (Stochastic Computation Graph)** *A stochastic computation graph (SCG) is a directed acyclic graph (DAG) $\mathcal{G} = (\mathcal{V}, \mathcal{E})$, where $\mathcal{V} = (\mathcal{I}, \mathcal{D}, \mathcal{S})$ is the set of nodes consisting of input nodes $\mathcal{I}$, deterministic nodes $\mathcal{D}$, and stochastic nodes $\mathcal{S}$, and $\mathcal{E}$ is the set of directed edges between nodes. In particular, input nodes are set externally, including input data, ground truth data and trainable parameters which are denoted specifically by $\Theta$; deterministic nodes are functions of their parents, including cost nodes which are scalar-valued functions denoted specifically by $\mathcal{C}$; and, stochastic nodes which are distributed conditionally on their parents.*

Given a SCG, it is assumed that all the cost nodes are deterministic. If a stochastic function is needed for a cost node, it can alternatively be constructed by attaching an identity function to a normal stochastic node. A node could either be a primitive computation, or a sophisticated composition of computations. In this paper, we view SCG in an abstract way, so a node is a set of computations with specific functionality at the high level. This abstraction is not needed to establish our main result, and the way to do abstraction over a SCG is beyond the scope of this paper.

Generally, there are two types of edges: differentiable and non-differentiable. An edge $(v, w)$ is differentiable if $w$ is deterministic and the Jacobian matrix $\frac{\partial w}{\partial v}$ exists, or $w$ is stochastic and $\frac{\partial}{\partial v} \Pr(w \mid \mathrm{PARENTS}_w)$ exists, where $\mathrm{PARENTS}_w$ is the set of parents of node $w$. Note that, if the partial derivative over an edge is intentionally ignored, then this edge is considered non-differentiable. For example, the derivatives with respect to the input and ground truth data are

usually ignored in practice. Two relations of influence are defined for a SCG. For any node $v$ and $w$ in the graph, if there exists a sequence of nodes $n_1, n_2, \ldots, n_K$, with $K \geq 1$, such that $(v, n_1), (n_1, n_2), \ldots, (n_K, w)$ are all edges in the graph, it is called that "$v$ influences $w$", denoted by $v \prec w$. If all the intermediate nodes $n_k$ (not including $v$ and $w$) are deterministic, it is called that "$v$ *deterministically* influences $w$", denoted by $v \prec^D w$.

Following the relation of deterministic influence, the set of dependencies of node $v$ is defined as the set of parameter and stochastic nodes that deterministically influence it, i.e. $\text{DEPS}_v := \{u \in \Theta \cup \mathcal{S} \mid u \prec^D v\}$. Input nodes other than parameters, such as input and ground truth data, are not considered as dependencies for other nodes, since they are usually considered as constants in one run of the graph and there is no need to compute derivatives with respect to them. If node $v$ is deterministic, the value of $v$ is a function of $\text{DEPS}_v$, we can write $v(\text{DEPS}_v)$, which also applies to cost nodes; if node $v$ is stochastic, the probability density/mass function of $v$ is a function of $\text{DEPS}_v$, we can write $\Pr(v \mid \text{DEPS}_v)$, in which case we use $\hat{v}$ to denote a sampled value of $v$. In the next section, it will be shown that this dependency relation suffices to estimate the derivatives of all cost nodes with respect to all trainable parameters, if a so-called *differentiable requirement* holds.

## 2.2 Gradient Estimation on Stochastic Computation Graph

Given a SCG $\mathcal{G}$ and a training dataset $D$, an optimization problem can be defined as finding the best set of parameter nodes $\Theta$ such that the expected sum of all cost nodes is minimized, more formally:

$$\underset{\Theta}{\text{minimize}} \; J(\Theta) = \mathbb{E}\left[\sum_{c \in \mathcal{C}} c(\text{DEPS}_c)\right]. \tag{1}$$

This optimization problem can be solved via standard stochastic gradient descent (SGD) if the gradient $\nabla J(\Theta)$ exists, in which case the whole graph is considered to be differentiable. It can be shown that this gradient exists, if the following differentiable requirement is met:

**Condition 1 (Differentiable Requirement)** *Given a SCG $\mathcal{G} = (\mathcal{V}, \mathcal{E})$, it is considered differentiable, if for any parameter node $\theta \in \Theta$, all edges $(v, w) \in \mathcal{E}$, satisfying $\theta \prec^D v$ and $\theta \prec^D w$, are differentiable: either $w$ is deterministic and Jacobian matrix $\frac{\partial w}{\partial v}$ exists; or $w$ is stochastic and $\frac{\partial}{\partial v}\Pr(w \mid \text{PARENTS}_w)$ exists.*

It is worth noting that this differentiable requirement does not require that all edges in the graph are differentiable. If the path from a parameter node $\theta$ to deterministic node $v$ is blocked by some stochastic nodes, then $v$ does not have to be differentiable with respect to its parents; if the path from a parameter node $\theta$ to a stochastic node $v$ is blocked by other stochastic nodes, then the probability density/mass function of $v$ conditioned on its parents does not have to be differentiable — in fact, it does not have to be known [1].

If a SCG is differentiable, then the partial derivative of $J(\Theta)$ with respect to any parameter node $\theta \in \Theta$ can be unbiasedly estimated as:

$$\frac{\partial}{\partial \theta} J(\Theta) = \mathbb{E}\left[\sum_{\theta \prec^D v, v \in \mathcal{S}} \left(\frac{\partial}{\partial \theta} \log \Pr(v \mid \text{DEPS}_v)\right)\hat{Q}_v + \sum_{\theta \prec^D c, c \in \mathcal{C}} \frac{\partial}{\partial \theta} c(\text{DEPS}_c)\right], \tag{2}$$

where $\hat{Q}_v = \sum_{c \in \mathcal{C}, v \prec c} \hat{c}$ is the sum of all sampled "downstream" costs influenced by stochastic node $v$. The two parts in Equation 2 are the derivatives originated from stochastic paths and deterministic paths respectively. If we let $L(\Theta) = \sum_{\theta \prec^D v, v \in \mathcal{S}} \log \Pr(v \mid \text{DEPS}_v)\hat{Q}_v + \sum_{\theta \prec^D c, c \in \mathcal{C}} c(\text{DEPS}_c)$ — a deterministic differentiable loss function, it can be seen that $\frac{\partial}{\partial \theta} J(\Theta) = \mathbb{E}\left[\frac{\partial}{\partial \theta} L(\Theta)\right]$. In other words, we can apply a standard automatic differentiation procedure to $L$ using sampled $\hat{Q}_v$ values to obtain an unbiased estimator of $\frac{\partial}{\partial \theta} J(\Theta)$. In this sense, $L$ is called the *surrogate* loss function, and the corresponding surrogate optimization problem is defined as:

$$\underset{\Theta}{\text{minimize}} \; L(\Theta) = \sum_{\theta \prec^D v, v \in \mathcal{S}} \log \Pr(v \mid \text{DEPS}_v)\hat{Q}_v + \sum_{\theta \prec^D c, c \in \mathcal{C}} c(\text{DEPS}_c). \tag{3}$$

---

[1]This is why reinforcement learning, as a specialization of SCG, can be solved model-free.

Solving the surrogate optimization problem defined in Equation 3 using standard SGD via back-propagation gives us an optimal set of parameters, which is also optimal for the original optimization problem defined in Equation 1. The surrogate loss function $L(\Theta)$, however, often has a high variance due to the sampled $\hat{Q}_v$ values, which may cause instability during training. To alleviate this issue, it is a common practice to subtract some "baseline" values $b$ from $\hat{Q}_v$ to reduce the variance without affecting the derivatives. As shown in (Greensmith et al., 2004; Schulman et al., 2015), taking $b = \mathbb{E}[\hat{Q}_v]$ is generally a good option.

## 3  REINFORCED PIPELINE OPTIMIZATION

In this paper, we assume a pipeline can be represented as a SCG, which is in turn a DAG by definition. A pipeline is differentiable if and only if the underlying SCG is differentiable, in which case it is automatically end-to-end trainable. In this section, we focus on pipelines that are not originally differentiable. For example, Figure 1 shows a Faster R-CNN pipeline in testing time, which is not differentiable due to the non-differentiable -mAP criterion and non-differentiable NMS and RoI pooling computations. We show that by applying a so-called stochastication trick on a non-differentiable pipeline, we obtain a stochastic but differentiable counterpart, which can then be trained end-to-end.

**Definition 2 (Stochastication)** *Given a non-differentiable pipeline represented as a stochastic computation graph $\mathcal{P} = (\mathcal{V}, \mathcal{E})$, let $\epsilon \subseteq \mathcal{E}$ be the set of non-differentiable edges that are not resulting from gradients being intentionally ignored. For an edge $(v, w) \in \epsilon$, if $v$ is deterministic, and there exists at least one parameter node $\theta \in \Theta$, such that $\theta \prec^D v$, we modify the pipeline by: 1) removing edge $(v, w)$ from $\mathcal{E}$, and 2) adding edges $(v, k)$ and $(k, w)$ into $\mathcal{E}$, where $k$ is a stochastic node with differentiable probability density/mass function $\Pr(k \mid v)$ such that $\mathbb{E}[k] = v$. Repeat this operation until no edges can be removed from or added to the pipeline. Return the new pipeline.*

For a non-differentiable pipeline $\mathcal{P} = (\mathcal{V}, \mathcal{E})$, let $\mathcal{P}' = (\mathcal{V}', \mathcal{E}')$ be the resulting pipeline after applying the stochastication trick on $\mathcal{P}$. Let $\epsilon \in \mathcal{E}$ be the set of edges belonging to $\mathcal{P}$ that are not originally differentiable. Let $\epsilon' \in \mathcal{E}'$ be the set of newly added edges belonging to $\mathcal{P}'$. For any parameter node $\theta \in \mathcal{V}'$, for any edge $(v, w) \in \mathcal{E}'$, satisfying $\theta \prec^D v$ and $\theta \prec^D w$, if $(v, w) \notin \epsilon'$, we must have $(v, w) \in \mathcal{E}$ and $(v, w) \notin \epsilon$, implying that $(v, w)$ is differentiable at the first place; on the other hand, if $(v, w) \in \epsilon'$, then $w$ must be the newly added stochastic node under the condition that $\theta \prec^D v$ and $\theta \prec^D w$, and $(v, w)$ must be differentiable because $\Pr(w \mid v)$ is differentiable by construction. In summary, the differentiable requirement is met for pipeline $\mathcal{P}'$, thus $\mathcal{P}'$ can be trained end-to-end via backpropagation in terms of the surrogate optimization problem defined in Equation 3, which solves the original optimization problem defined in Equation 1. Figure 3 gives the converted pipeline for end-to-end training on Faster R-CNN. The only structural difference between Figure 3 and Figure 1 is the two newly added stochastic nodes $\pi_0$ and $\pi_1$ — which closes the gap between training and testing while being able to optimize on the real criterion -mAP.

In training of $\mathcal{P}'$ as solving the surrogate optimization problem, once the training is converged, it is induced that we have an optimal pipeline with a set of optimal parameters $\Theta^*$ over the training dataset (globally or locally optimal depending only on the training process not the pipeline itself). We now show that the original pipeline $\mathcal{P}$ equipped with the same set of learned parameters $\Theta^*$ is also optimal over the same training dataset under certain conditions.

Recall that, in the converted pipeline $\mathcal{P}' = (\mathcal{V}', \mathcal{E}')$, the objective function is defined as $J'(\Theta) = \mathbb{E}\left[\sum_{c \in \mathcal{C}} c(\text{DEPS}_c)\right]$. Let $\kappa = \mathcal{V}' - \mathcal{V}$ be the set of newly added stochastic nodes, and $\kappa_c = \{k \in \kappa \mid k \prec^D c\}$ be the set of stochastic nodes within $\kappa$ influencing cost node $c$. We can decompose $J'$ into two parts: $J'(\Theta) = \mathbb{E}\left[\sum_{c \in \mathcal{C}, \kappa_c \neq \varnothing} c(\text{DEPS}_c)\right] + \mathbb{E}\left[\sum_{c \in \mathcal{C}, \kappa_c = \varnothing} c(\text{DEPS}_c)\right]$. The second part is not affected by the newly added stochastic nodes. For the first part, we have $\mathbb{E}\left[\sum_{c \in \mathcal{C}, \kappa_c \neq \varnothing} c(\text{DEPS}_c)\right] = \mathbb{E}\left[\sum_{c \in \mathcal{C}, \kappa_c \neq \varnothing} c(\kappa_c, \text{DEPS}_c - \kappa_c)\right]$. Noting that for each node $k \in \kappa_c$, we have $\mathbb{E}[k] = v$ where $v$ is the preceding deterministic node of $k$, we can then write $\hat{k} = v + z_k$ such that $\hat{k}$ is a sampled value of $k$, and $z_k$ is a zero-mean random noise term. It follows that $c(\kappa_c, \text{DEPS}_c - \kappa_c) = c(\text{PARENTS}_{\kappa_c}, \text{DEPS}_c - \kappa_c) + z_c$ where $\text{PARENTS}_{\kappa_c} = \{v_1, v_2, \ldots, v_K\}$ is the set of preceding deterministic nodes of $\kappa_c$, and $z_c$ is the amortized noise term depending on

1  **RPO** ($\mathcal{P}$ : *pipeline*, $D$ : *training dataset*, $N$ : *number of iterations*)
2  Let $\mathcal{P}' \leftarrow$ **Stochastication** ($\mathcal{P}$)
3  **for** $i \leftarrow 1$ **to** $N$ **do**
4     Sample random mini-batch from $D$
5     Run $\mathcal{P}'$ once without sampling:
6     - Collect baseline values $b_k$ for newly added stochastic node $k \in \kappa$
7     - Collect sum of differentiable costs (if any) $s = \sum_{c \in \mathcal{C}, \exists \theta \in \Theta : \theta \prec^D c} c$
8     Run $\mathcal{P}'$ once with sampling:
9     - Collect log probabilities $\{\log \Pr(k \mid \mathrm{DEPS}_k)\}$ for $k \in \kappa$
10     - Collect sums of influenced costs $\{\hat{Q}_k\}$ for $k \in \kappa$
11     Construct a surrogate loss $L = s + \sum_{k \in \kappa} \log \Pr(k \mid \mathrm{DEPS}_k) \left( \hat{Q}_k - b_k \right)$
12     Compute gradient $\nabla_\Theta L$ via backpropagation
13     Update parameters $\Theta$ with $\nabla_\Theta L$
14  **return** pipeline $\mathcal{P}$ with learned parameters $\Theta$

Figure 4: Reinforced pipeline optimization (RPO) with baseline. Note that the original pipeline may have some additional differentiable costs (line 7), which are optimized simultaneously (line 11). The original pipeline is assumed to only have deterministic nodes. If it in fact has some stochastic nodes, only slight modifications are needed for extension.

$v_1, v_2, \ldots, v_K$ and $z_{k_1}, z_{k_2}, \ldots, z_{k_K}$. Then under the condition that $\mathbb{E}[z_c]$ is a constant, we have $\mathbb{E}\left[ \sum_{c \in \mathcal{C}, \kappa_c \neq \varnothing} c(\mathrm{DEPS}_c) \right] = \sum_{c \in \mathcal{C}, \kappa_c \neq \varnothing} c(\mathrm{PARENTS}_{\kappa_c}, \mathrm{DEPS}_c - \kappa_c) + \sum_{c \in \mathcal{C}, \kappa_c \neq \varnothing} \mathbb{E}[z_c]$. In other words, the first part can be emerged into the second part. More precisely, $J'(\Theta) = \sum_{c \in \mathcal{C}, \kappa_c \neq \varnothing} c(\mathrm{PARENTS}_{\kappa_c}, \mathrm{DEPS}_c - \kappa_c) + \mathbb{E}\left[ \sum_{c \in \mathcal{C}, \kappa_c = \varnothing} c(\mathrm{DEPS}_c) \right] + \sum_{c \in \mathcal{C}, \kappa_c \neq \varnothing} \mathbb{E}[z_c] = J(\Theta) + C$, where $C$ is a constant. Therefore, under a so-called *constant drift* condition, we have $J'(\Theta) = J(\Theta) + C$ for any set of parameters $\Theta$. It is then clear that, if $\Theta^*$ is optimal for $J'$ over a training dataset, it is also optimal for $J$ over the same training dataset.

**Condition 2 (Constant Drift Requirement)** *Let $\kappa = \mathcal{V}' - \mathcal{V}$ be the set of newly added stochastic nodes. Let $\kappa_c = \{k \in \kappa \mid k \prec^D c\}$ be the set of stochastic nodes within $\kappa$ influencing cost node $c$. Let $c(\kappa_c, \mathrm{DEPS}_c - \kappa_c) = c(\mathrm{PARENTS}_{\kappa_c}, \mathrm{DEPS}_c - \kappa_c) + z_c$, where $z_c$ is the amortized noise term. If the noise term $z_c$ has a constant mean, the converted stochastic pipeline and the original pipeline share the same set of optimal parameters over the same training dataset.*

In the case when all cost nodes $c$, influenced by stochastication, are linear with respect to the newly added stochastic nodes, it is easy to show that the constant drift condition is met naturally (in which case the noise term has a zero mean). Generally, it is difficult to obtain a closed-form expression for nonlinear cost functions of several random variables to check whether the constant drift condition is met. However, if the random variables are jointly Gaussian with nonzero cross-correlation, then it is possible to compute useful expectations when the nonlinear functions are memoryless (Shynk, 2012), which is beyond the scope of this paper.

In order to reduce variance in training of $\mathcal{P}'$, it suffices to simply use the expected value of $\hat{Q}_k$ for each stochastic node $k \in \kappa$. Having $\mathbb{E}[k] = v$, where $v$ is the preceding deterministic node of $k$, it follows that the baseline values can be collected by running the pipeline $\mathcal{P}'$ once without sampling (which is equivalent to running the original pipeline $\mathcal{P}$). Figure 4 outlines the main algorithm. We call this algorithm *reinforced pipeline optimization* (RPO), since it is very similar to *policy gradient* (Sutton et al., 2000) in the context of *reinforcement learning* (RL), except that RL relies on a Markovian model which has to be a linear stochastic computation graph in the dimension of time (Sutton et al., 1998), but RPO is generally applicable to any stochastic computation graph that can be represented as a DAG. RPO in Figure 4 adopts an implementation with baseline. If needed it is rather straightforward to discard baseline from it. In experiments, we test RPO with and without baseline for ablation study.

It can be interpreted that RPO solves an optimization problem defined over a stochastic computation graph via stochastic gradient descent using standard backpropagation, which happens to find the best set of parameters for the original pipeline under the condition of constant drift. If the underlying pipeline can be represented as a linear graph (i.e. a chain of nodes), it becomes a *Markov decision process* (MDP) where the information flow passing through the pipeline is the sequence of states, trainable models are parts of the policy, alternations made to states are actions, the performance criterion is the reward function being optimized, and other non-trainable parts are parts of the environment. Under this MDP formulation, RPO reduces to standard policy gradient. The optimal policy of the resulting MDP corresponds to the optimal stochastic pipeline possible.

One concern on the proposed algorithm is that it might learn slowly if trained from scratch since it relies solely on the final criterion (such as -mAP for a Faster R-CNN pipeline) which is usually sparse and cannot provide relatively strong signals for the learning process. In practice, we can always supply additional cost/reward signals to guide the learning process, using techniques such as *reward shaping* (Ng et al., 1999). In this sense, isolated loss functions developed for traditional non-optimal pipeline training can be reused here for accelerating end-to-end training in the early stage of training without affecting the optimality result we established. In the case the criterion function has some internal structure, we can also apply a so-called *reward decomposition* method on it to explicitly exploit the structure in order to obtain decomposed problems that are easier to learn (Dieterich, 2000; Russell & Zimdars, 2003; Marthi, 2007). In our default experiment setting, we adopt pretrained models before applying end-to-end reinforced training.

## 4 EMPIRICAL EVALUATION

We conduct our main experiments on Faster R-CNN which is a state-of-the-art object detection pipeline (Ren et al., 2015). We choose Faster R-CNN for several reasons. As shown in Figure 1, Faster R-CNN has a clear two-stage structure, namely the region proposal network (RPN) and the region convolutional neural network (R-CNN). These two stages are not linearly combined, as they share the same base feature generated from one backbone network (which prevents it from being simply modeled as an MDP). The final criterion measuring its real performance, i.e. the mAP value, is well-known to be non-differentiable. In addition, Faster R-CNN contains many other non-differentiable computations, such as RoI pooling and NMS. Furthermore, object detection is a non-trivial one-to-many problem, in the sense that, given a single image input (fixed size or not), there are potentially many output entities without knowing the exact number in advance. Therefore, end-to-end optimizing an object detection pipeline, such as Faster R-CNN, is a challenging problem for pipeline optimization.

We now describe in detail how we convert a Faster R-CNN pipeline into a stochastic counterpart and train it end-to-end using the proposed RPO algorithm. In the first stage, the raw output of RPN for each image within a mini-batch is a set of bounding boxes $B_{\text{RPN}} = \{b\}$ where a bounding box $b$ is represented as $b = (p_b, o_b)$. Here, $p_b$ is a softmax score giving the probability of bounding box $b$ being an object, and $o_b = (t_x, t_y, t_w, t_h)$ is the bounding box offset with respect to an anchor box $a$, defined as $t_x = \frac{x-x_a}{w_a}$, $t_y = \frac{y-y_a}{h_a}$, $t_w = \log(\frac{w}{w_a})$, and $t_h = \log(\frac{h}{h_a})$, where $x$ and $x_a$ are the x-coordinates for the predicted box and anchor box respectively (likewise for y-coordinate $y$, width $w$ and height $h$). Following Ren et al. (2015), we ignore all cross-boundary bounding boxes. After applying stochastication, during sampling, we firstly construct a Bernoulli distribution for each bounding box score with $p_b$ as the mean, denoted by $\mathcal{B}(p \mid p_b)$, from which we sample a 0-1 score $\hat{p}_b \sim \mathcal{B}(p \mid p_b)$. We then construct a multivariate Gaussian distribution for each bounding box offset with $o_b = (t_x, t_y, t_w, t_h)$ as the mean and a constant matrix $\boldsymbol{\Sigma} = \sigma^2 \boldsymbol{I}$ as the covariance, denoted by $\mathcal{N}(o \mid o_b, \boldsymbol{\Sigma})$. Here, $\boldsymbol{I}$ is an identity matrix and $\sigma$ is a hyper parameter which needs to be set before training. For each bounding box offset, we then sample a new offset nearby $\hat{o}_b \sim \mathcal{N}(o \mid o_b, \boldsymbol{\Sigma})$. By sampling $\hat{p}_b$ and $\hat{o}_b$, we obtain a new bounding box $\hat{b} = (\hat{p}_b, \hat{o}_b)$. The overall log probability for $\hat{b}$ is computed as $\log \Pr(\hat{b} \mid b) = \log \mathcal{B}(\hat{p}_b \mid p_b) + \hat{p}_b \log \mathcal{N}(\hat{o}_b \mid o_b, \boldsymbol{\Sigma})$, implying that if a bounding box is sampled as background (i.e. $\hat{p}_b = 0$) then its offset-part log probability is ignored. We then accumulate and average log probabilities for all bounding boxes within the same image, which reads $\frac{1}{|B_{\text{RPN}}|} \sum_{b \in B_{\text{RPN}}} \log \Pr(\hat{b} \mid b)$. The set of newly sampled bounding boxes is then passed to following stages of the pipeline. This step is depicted as node $\pi_0$ in Figure 3.

| Backbone Network | ResNet-18 | ResNet-34 | ResNet-101 | VGG-16 |
|---|---|---|---|---|
| Pre-trained | 50.57% | 61.24% | 74.89% | 66.69% |
| RPO w/ baseline | $52.25 \pm 0.07\%$ | $61.94 \pm 0.12\%$ | $75.05 \pm 0.10\%$ | $67.44 \pm 0.11\%$ |
| RPO w/o baseline | $52.25 \pm 0.05\%$ | $61.84 \pm 0.05\%$ | $75.04 \pm 0.07\%$ | $67.43 \pm 0.10\%$ |

Table 1: Empirical results shown as mAP values on PASCAL VOC 2007 validation set after training on training set. The mAP Values are obtained before and after applying RPO with and without baseline over various backbone networks.

Similarly, in the second stage, the raw output of R-CNN for each image is a set of bounding boxes $B_{\text{RCNN}} = \{b\}$. But here $b$ is represented as $b = (\boldsymbol{p}_b, \boldsymbol{r}_b)$, where $\boldsymbol{p}_b$ is a vector of softmax probabilities with $\boldsymbol{p}_b[c]$ being the probability of bounding box $b$ belonging to class $c$, and $\boldsymbol{r}_b$ is a vector of rectangles with $\boldsymbol{r}_b[c] = (x_0, y_0, x_1, y_1)$ being the predicted rectangle conditioned that $b$ belongs to class $c$. During sampling, we firstly construct a categorical distribution for each bounding box with $\boldsymbol{p}_b$ as the mean, denoted by $\mathcal{C}(\boldsymbol{p} \mid \boldsymbol{p}_b)$, from which we sample a one-hot distribution $\hat{\boldsymbol{p}}_b \sim \mathcal{C}(\boldsymbol{p} \mid \boldsymbol{p}_b)$. We then construct a vector of multivariate Gaussian distributions for each bounding box with $\boldsymbol{r}_b[c]$ as the means, denoted by $\mathcal{N}(r \mid \boldsymbol{r}_b[c], \boldsymbol{\Sigma})$ where $\boldsymbol{\Sigma} = \sigma^2 \boldsymbol{I}$, from which we sample a set of new rectangles nearby $\hat{r}_b[c] \sim \mathcal{N}(r \mid \boldsymbol{r}_b[c], \boldsymbol{\Sigma})$. At this point, we obtain a new bounding box $\hat{b} = (\hat{\boldsymbol{p}}_b, \hat{\boldsymbol{r}}_b)$. Let the vector of log probabilities over all classes be $\log \Pr(\hat{\boldsymbol{r}}_b \mid \boldsymbol{r}_b) = [\log \mathcal{N}(\hat{\boldsymbol{r}}_b[c] \mid \boldsymbol{r}_b[c], \boldsymbol{\Sigma})]$. The overall log probability is computed as $\log \Pr(\hat{b} \mid b) = \log \mathcal{C}(\hat{\boldsymbol{p}}_b \mid \boldsymbol{p}_b) + (1 - \hat{\boldsymbol{p}}_b[0]) \log \Pr(\hat{\boldsymbol{r}}_b \mid \boldsymbol{r}_b) [\arg\max_c \hat{\boldsymbol{p}}_b[c]]$. The second term indicates that only the log probability corresponding to a sampled non-background class is considered. We then accumulate and average log probabilities for all bounding boxes within the same image, which reads $\frac{1}{|B_{\text{RCNN}}|} \sum_{b \in B_{\text{RCNN}}} \log \Pr(\hat{b} \mid b)$. Finally, the set of newly sampled bounding boxes is passed to the NMS operation before computing -mAP. This step is depicted as node $\pi_1$ in Figure 3.

We implement our algorithm based on a publicly available Faster R-CNN repository [2]. We evaluate the proposed RPO algorithm on the widely-used PASCAL VOC 2007 benchmark (Everingham et al., 2008), which contains about 5k trainval images and 5k test images over 20 object categories. By default in our implementation, we firstly train Faster R-CNN with various backbone networks following a normal way with a learning rate of 0.001 which is decreased every 15 epochs with a factor of 0.1 until no further improvements can be achieved; we then apply RPO with and without baseline on the pre-trained models for additional 15 epochs. Throughout reinforced training, we use a standard SGD optimizer with a momentum of 0.9 and a learning rate of 0.0001, which is decayed every 5 epochs with a factor of 0.9. The hyper parameter for the Gaussian distributions is set to be $\sigma = 1$. Typically, a traditional Faster R-CNN pipeline may use different parameters for training and testing. For example, in the original implementation, the authors set the number of proposal regions to be 2,000 in training time, and 300 in testing time. In our implementation, we use exactly the same parameters for both training and testing time. For instance, we set the number of proposal regions to be 300 in both time. This can be seen as one advantage of our proposed method, since it further closes the gap between training and testing.

In reinforced training of Faster R-CNN, the final -mAP value is considered as the only cost node. The collected log probabilities multiplied by the downstream -mAP value are used to construct the surrogate loss, to which we conduct an automatic differentiation procedure via backpropagation. In the case baseline is used, the -mAP value corresponding to a non-sampling pipeline is considered as a baseline, and will be subtracted from the -mAP value resulting from sampling for variance reduction. We notice it is generally difficult to justify whether the constant drift condition is met in this setting, since the mAP value involves many non-linear computations over many random variables. However, empirically it is reasonable to assume that the learned parameters are near-optimal for the original Faster R-CNN pipeline.

Table 1 summaries experimental results in terms of mAP values before and after applying RPO averaged over 5 runs. Figure 5 reports more detailed results throughout training. Figure 6 presents some detection results of Faster R-CNN with ResNet-18 before and after applying RPO. It can be seen

---

[2]https://github.com/jwyang/faster-rcnn.pytorch

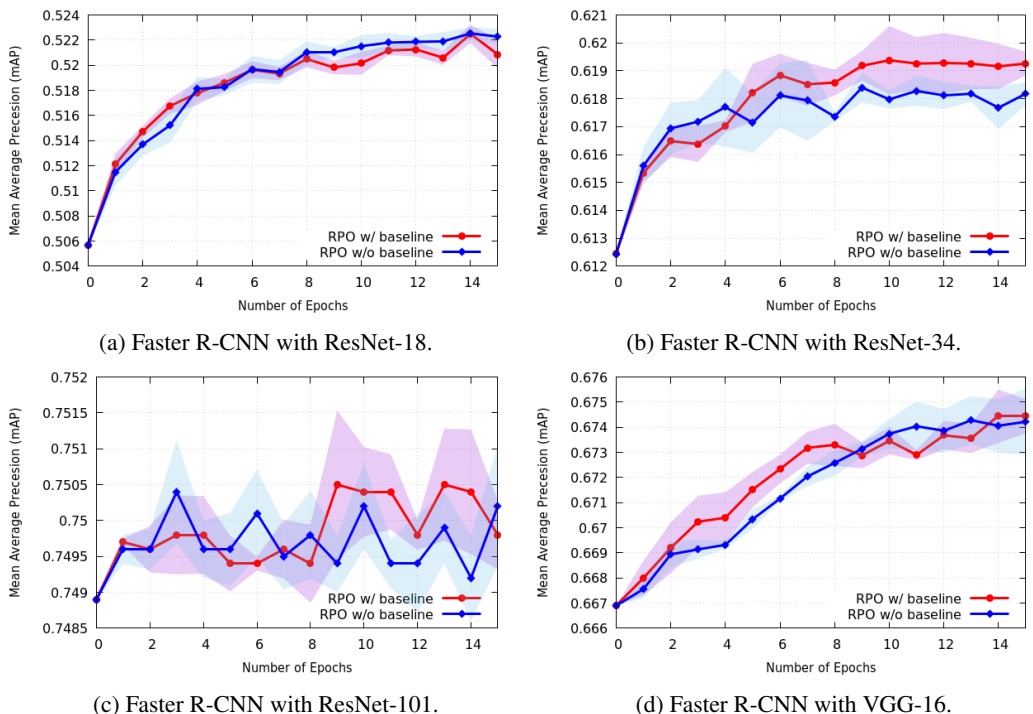

(a) Faster R-CNN with ResNet-18.

(b) Faster R-CNN with ResNet-34.

(c) Faster R-CNN with ResNet-101.

(d) Faster R-CNN with VGG-16.

Figure 5: Experimental results shown as mAP values on PASCAL VOC 2007 validation set while training on training set. The results are averaged over 5 runs of RPO with and without baseline on Faster R-CNN with various backbone networks. The $x$- and $y$- axes are number of epochs and mAP value respectively. The shaded area represents 95% confidence interval.

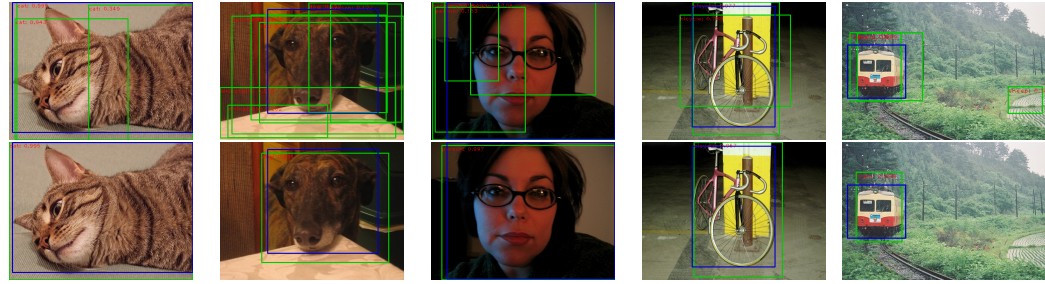

Figure 6: Selected detection results of Faster R-CNN with ResNet-18 before (top row) and after (bottom row) applying RPO. Blue boxes are ground truths; green boxes are predictions. These test images are from PASCAL VOC 2007 validation set. It can be seen that the results after reinforced training make much more sense.

from these results that RPO consistently improves a pre-trained pipeline with performance boost over ResNet-18, ResNet-34, ResNet-101 (He et al., 2016) and VGG-16 (Simonyan & Zisserman, 2014) backbone networks. The results also show that RPO improves a light model significantly, but not for a heavy model (such as ResNet-101). This might be because that a heavy model has relatively large capacity, such that it has leaned sufficient amount of knowledge and is already near-optimal during pre-training. As a result, it cannot be easily improved further in reinforced training. As ablation study, we test RPO with and without baseline. It can also be seen from the results that there are only slight differences between these two versions of RPO, suggesting that the baseline value we used may not be necessary for optimizing Faster R-CNN.

## 5 RELATED WORK

Caicedo & Lazebnik (2015) and Mathe et al. (2016) propose novel methods that learn to localize objects in scenes using an attention-action strategy, which is further extended to multiple objects in (Kong et al., 2017). Their methods model visual object detection as a sequential search problem (including stopping condition), and develop a reinforcement learning algorithm to approximately solve this problem. Their sequential models learn to predict detection hypothesis one at a time, until a stopping condition is generated, while in our method the joint policy, i.e. the two-stage networks, is able to generate all possible detection hypotheses concurrently. Very recently, Xie et al. (2018) develop a reinforcement learning algorithm to optimize a non-differentiable two-stage pipeline in inference time. The idea is to train a policy network to iteratively refine the output of the first stage in a way to make it more favorable to the second stage by optimizing some criteria. Their method needs to introduce an additional policy network, while our method treats the trainable models in the original pipeline as policies directly. Besides, their method runs over multiple iterations in the inference time, while our method runs completely in the training time, and keeps the original pipeline untouched in the inference time, which is much more efficient. Another recent related work is (Pirinen & Sminchisescu, 2018), where the authors train a reinforcement learning agent to replace the greedy RoI selection process with a sequential attention mechanism. Similar to (Xie et al., 2018), they introduce a separate sequential network to represent the policy, while in our method learning models are trained as policies directly. Rao et al. (2018) develop a policy gradient algorithm for learning a globally optimal Faster R-CNN object detector. Their method can be seen as an incomplete implementation of our method, in the sense that: 1) they sample a set of bounding boxes according to the softmax scores, but not the coordinates; and 2) they sample in the second stage, but not in the first stage. Besides, since bounding box coordinates are ignored in their sampling strategy, their method still relies on the standard R-CNN regression loss for training bounding box regressor. In contrast, our method optimizes a pipeline directly and solely on the final criterion that measures its real performance.

## 6 CONCLUSION

In this paper, according to the theory of stochastic computation graph, we show that by converting an originally non-differentiable pipeline into a stochastic counterpart, we can then train the converted pipeline end-to-end and optimize it with any criterion attached to it. We develop a stochastication trick, and propose a novel algorithm for end-to-end pipeline optimization, namely reinforced pipeline optimization (RPO). We further show that under a constant drift condition, the converted stochastic pipeline and the original pipeline share the same set of optimal parameters over the same training dataset. In experiments, we apply RPO to the optimization of challenging Faster R-CNN pipelines, and obtain empirically near-optimal object detectors consistent with Faster R-CNN design in terms of mean average precision. In future work, we would like to extend RPO to more general pipelines that are not necessarily DAGs, for example dynamic pipelines where some computations may be dependent on some other past computations in time.

ACKNOWLEDGMENTS

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
