# OpenReview forum: "Reinforced Pipeline Optimization: Behaving Optimally with Non-Differentiabilities"
_ICLR.cc/2019/Conference_

### Official Review · AnonReviewer3 · 2018-11-01
**Lacking meaningful baselines and some claims are dubious**

**Rating:** 3
**Confidence:** 4

**Review:**

Pros:
+ Improving joint training of non-differentiable pipelines is a meaningful and relevant problem
+ Using the stochastic computation graph structure to smooth a pipeline in a structured way is a plausible idea

Cons:
+ The main result of the paper concerning sufficient conditions for optimality of the method seems dubious
+ It is not obvious why this method would outperform simple baselines, and baselines for joint training were tried
+ The notation seems unnecessarily bloated and overly formal
+ The exposition spends too much time on prior work, too little on the contribution, and the description of the contribution is confusing

The submission describes a method for smoothing a non-differentiable machine learning pipeline (such as the Faster-RCNN detector), so that gradient-based methods may be applied to jointly train all the parameters of the pipeline.  In particular, the proposal involves recasting the pipeline as a stochastic computation graph (SCG), adding stochastic nodes to this graph, and then using REINFORCE-style policy gradients to perform parameter learning on the SCG.  It is claimed that under certain conditions, the optimal parameters of the resulting SCG are also optimal for the original pipeline.  The method is applied to optimizing the parameters of Faster-RCNN.

I think making non-differentiable pipelines differentiable is an intuitively appealing concept.  A lot of important, practical machine learning systems fall into this category, so devising a nice way to do global parameter optimization for such systems could potentially have significant impact.  In general, we can’t hope to make much meaningful progress on the problem of optimizing general nonlinear, differentiable functions, but it is plausible that a method that targets key non-differentiable components for smoothing—such as this paper—could outperform a generic black-box optimizer.  So, I think the basic idea here is plausible and addresses an important problem.

Unfortunately, I think this work loses sight of that high-level goal: to me, the key question is whether the proposed approach outperforms any other simple method for global parameter optimization in the presence of nonlinearities and nondifferentiability.  The paper fails to answer this question because no baselines for global parameter optimization were tried.  We can just treat the pipeline as a black box mapping parameters to training set performance, and so any black-box optimization method can be applied to this problem.  It is not clear that the proposed method would outperform an arbitrary black box optimization method such as simulated annealing, Nelder-Mead, cross-entropy method, etc.

I think there are also much simpler methods in a similar vein to the proposed method that might also perform just as well as the proposal.  One key conceptual issue here is that reducing the problem to a reinforcement learning problem, as the submission does, is not much of a reduction at all.  First, if the goal is to do global parameter optimization, then we don’t really have to smooth the pipeline itself: we can just smooth the black box mapping parameters to performance, and then optimize that with SGD.  There are many ways to do this--if we want to use policy gradient, we can just express the problem as something in this form:

min_\phi E_{\theta ~ q_\phi} C(\theta)

where C is the black-box mapping parameters \theta to a performance index (such as mean AP), q_\phi is a distribution over parameters (e.g., Gaussian), and \phi are the distribution parameters (e.g., mean, covariance of the Gaussian).  We can then optimize this using REINFORCE policy gradients.

If we want to really smooth the pipeline itself, then it is also easy to do this by devising a suitable MDP and then applying REINFORCE with the usual MDP structure.  We simply identify the state s_t at time t with the output of the t’th pipeline stage, introduce a new ‘action’ variable a_t representing a ’stochastified output’, and trivial dynamics (P(s_{t+1} | s_t, a_t) = \delta(s_{t+1} - a_t)).  If the policy is a Gaussian (P(a_t | s_t) = N(a_t; s_t, \Sigma)), then this is similar to relaxing the constraint that one stage’s output is equal to the input of the next stage, and somehow quadratically penalizing their difference.  In fact, there is a neural network training method based explicitly on this penalization view [A], and it would make yet another great baseline to try.

In fact, the proposed method is essentially similar to what I have just described, but it is unfortunately described in an overcomplicated way that obscures the true nature of the method.  I think the whole SCG framework is overkill here.  Too much of the paper is spent just rehashing the SCG framework, and the very heavy notation again just obscures the essential character of the method.

If there were, as the paper claims, some interesting condition under which the method produces solutions that are optimal under the original pipeline, that would be remarkable and interesting.  However, I have serious doubts about this part of the paper.  The key problem is the statement that “It follows that c(k_c, DEPS_c - k_c) = c(…) + z_c”.  The paper seems to be claiming that if E z = 0, then c(k + z) = c(k) + z, which can’t possibly be true in general.

The heavy and opaque notation makes it very difficult to understand this section.  Perhaps it would help to consider a very simple example.  Suppose we want to minimize E_{x ~ q} c(y(x)) (where x ~ q means x is distributed as q).  We can introduce only one new stochastic node (k = y + z), between y and c.  Clearly c(y + z) is not generally equal to c(y) + z, even if E z = 0.

In summary, I think the submission needs a lot of work on multiple axes before it can make a significant impact.  The most important issues are a complete lack of relevant baselines and the dubious claims about sufficient conditions for optimality.  The idea could have merit, but it needs to be carefully compared and motivated with respect to existing work (such as [A]) as well as the simple baselines I have mentioned.  The presentation also needs to be revised to find the simplest expression of the method and to focus on the interesting parts.

[A] Taylor, Gavin, et al. "Training neural networks without gradients: A scalable admm approach." International Conference on Machine Learning. 2016.

---

### Official Review · AnonReviewer2 · 2018-11-02
**Unclear about the extent of the contribution**

**Rating:** 5
**Confidence:** 2

**Review:**

The paper proposes a method for converting a non-differentiable machine learning pipeline into a stochastic, differentiable, pipeline that can be trained end-to-end with gradient descent approaches.

* Clarity: The language in the paper is very clear and easy to follow. The paper is lacking in clarity only when discussing some results/concepts from previous work (see detailed comments below).
* Quality: Overall the paper is in good shape, aside from some concerns which I will describe further.
* Originality: The originality is not very clear because it seems that a lot of ideas are borrowed from Schulman et al. (2015) (i.e. the concept of stochastic computation graph and how to compute the gradient) and from Rao et, al (2018) (i.e. sampling bounding boxes in some stages of the pipeline). To be fair to the authors, I am not very familiar with the two papers mentioned above, which makes this hard to judge. However, I think this paper could have explained more clearly which part exactly is a novelty of this paper, and where it separates from the rest.
* Significance: The concept of converting a non-differentiable pipeline to a differentiable version is indeed very useful and widely applicable, but the experimental section did not convince me that this particular method indeed works: the results show a very small improvement (0.7-2%) on a single system (Faster R-CNN), that has already been pretrained (so not clear if this method can learn from scratch).

Pros:
1)	Overall the paper is well written.
2)	The algorithm shown in Figure 4 nicely summarizes the whole algorithm.
3)	I particularly liked the part of Section 3 where it is shown the equivalence between the optimal parameters for the non-differentiable pipeline and the optimal parameters for the differentiable version.
4)	Figure 5 with detailed results is useful.

Cons:
5)	The way the paper is written, it is not clear where the contribution of this paper separates from existing work, mainly Schulman et al. (2015). I believe the idea of going around non-differentiability via minimizing a surrogate loss (i.e. your equation (2) introduced by Schulman et al. (2015)) is already known. I’m not sure exactly where this work diverges from that.
6)	The contribution of this paper is posed as a general framework for turning an arbitrary non-differentiable pipeline into a similar differentiable and stochastic version. However, the experimental section does not convince me that:
    a)	it is general – because it is applied only on the Faster R-CNN problem.
    b)	that it can learn from scratch – it is only applied after the base method has been pre-trained. There are no experiments where you train a network from scratch with this new differentiable pipeline. If the reason is that ResNets are hard to train from scratch, then you can always try your pipeline on a smaller problem, even a synthetic dataset, just to prove that it works.
     c)	that the improvement is significant from the baseline method – the results section show only a 1-2% increase in mAP, and only for the smaller networks (on larger ResNet models the gain is less than 1%, and the standard deviation is getting larger).

Detailed comments:
7)	You only cite the work of Schulman et al. (2015) at the beginning of section 2.1. While moving to section 2.2, I initially got the wrong impression that this us your contribution. Please state clearly where this comes from.
8)	It is not explained well why the new gradient can be estimated as in equation (2). I spent quite some time trying to figure out where that comes from (particularly the log part), only to realize that the explanation is probably in the original work (at the time when I thought this is your contribution). Please point the readers to it.

Final remarks:
Overall this paper introduces some interesting ideas. My main concerns were: (1) the originality, and (2) the results are not convincing. Perhaps concern (1) can be easily clarified by the authors, but for concern (2) it might be useful to show new results (training from scratch, other architectures to prove generality), as well as give arguments as to why the 1-2% gain in mAP is significant.

---

### Official Review · AnonReviewer1 · 2018-11-02
**paper review**

**Rating:** 4
**Confidence:** 5

**Review:**

The authors use RPO (Shulman et al, 2015) to transform non-differentiable operations in Faster R-CNN such as NNS, RoIPool, mAP to stochastic but differentiable operations. They cast Faster R-CNN as a SCG which can be trained end-to-end. They show results on VOC 2007.

Pros:
(+) The idea of casting a non-differentiable pipeline into a stochastic one is very reasonable
(+) This idea is showcased for a hard task, rather than toy examples, thus making it more realistic and exciting
Cons:
(-) Results are rather underwhelming
(-) Important properties of the final approach, such as complexity (time, memory, FLOPs) are not mentioned at all

While the idea the authors present seems reasonable and is showcased for a hard problem, such as object detection and on a well-designed system such as Faster R-CNN, the results are rather underwhelming. The proposed approach does not show any significant gains on top of the original pipeline (for ResNet101 the reported gains are < 0.2%). These small gains come at the expense of a more complicated definition and training procedure. The added complexity is not mentioned by the authors, such as time, memory requirements and FLOPs. In addition, the VOC2007 benchmark is rather outdated and much smaller than others. It would be nice to see similar results on COCO, which is larger and more challenging.

Similar efforts in this direction, namely making various modules of the Faster R-CNN pipeline differentiable, have shown little gains as well. For example, Dai at al., CVPR 2016, convert RoIPool into RoIWarp (following STN, Jaderberg et al) that allows for differentiation with respect to the box coordinates.

---

### Meta-Review · Area_Chair1 · 2018-12-14
**No rebuttal submitted**

**Confidence:** 5
**Recommendation:** Reject

**Metareview:**

The work proposes a method for smoothing a non-differentiable machine learning pipeline (such as the Faster-RCNN detector) using policy gradient. Unfortunately, the reviewers identified a number of critical issues, including no significant improvement beyond existing works. The authors did not provide a rebuttal for these critical issues.